# Machine Learning-Based Predicted Age of the Elderly on the Instrumented Timed Up and Go Test and Six-Minute Walk Test

**DOI:** 10.3390/s22165957

**Published:** 2022-08-09

**Authors:** Jeong Bae Ko, Jae Soo Hong, Young Sub Shin, Kwang Bok Kim

**Affiliations:** Digital Healthcare R&D Department, Korea Institute of Industrial Technology, Cheonan 31056, Korea

**Keywords:** dynamic balance ability, inertial measurement unit (IMU), XAI, timed up and go test (TUG), six-minute walk test

## Abstract

A decrease in dynamic balance ability (DBA) in the elderly is closely associated with aging. Various studies have investigated different methods to quantify the DBA in the elderly through DBA evaluation methods such as the timed up and go test (TUG) and the six-minute walk test (6MWT), applying the G-Walk wearable system. However, these methods have generally been difficult for the elderly to intuitively understand. The goal of this study was thus to generate a regression model based on machine learning (ML) to predict the age of the elderly as a familiar indicator. The model was based on inertial measurement unit (IMU) data as part of the DBA evaluation, and the performance of the model was comparatively analyzed with respect to age prediction based on the IMU data of the TUG test and the 6MWT. The DBA evaluation used the TUG test and the 6MWT performed by 136 elderly participants. When performing the TUG test and the 6MWT, a single IMU was attached to the second lumbar spine of the participant, and the three-dimensional linear acceleration and gyroscope data were collected. The features used in the ML-based regression model included the gait symmetry parameters and the harmonic ratio applied in quantifying the DBA, in addition to the features of description statistics for IMU signals. The feature set was differentiated between the TUG test and the 6MWT, and the performance of the regression model was comparatively analyzed based on the feature sets. The XGBoost algorithm was used to train the regression model. Comparison of the regression model performance according to the TUG test and 6MWT feature sets showed that the performance was best for the model using all features of the TUG test and the 6MWT. This indicated that the evaluation of DBA in the elderly should apply the TUG test and the 6MWT concomitantly for more accurate predictions. The findings in this study provide basic data for the development of a DBA monitoring system for the elderly.

## 1. Introduction

With the rapid progression of the aging population worldwide, the ability of elderly individuals to perform the activities of daily living (ADL) is increasingly recognized as a critical factor in maintaining a healthy life in the elderly. ADLs refer to the basic activities required to maintain daily living, such as eating, bathing and personal hygiene, dressing, continence, toileting, and walking and transferring [1]. A decline in the performance of ADLs can disturb independent daily living of the elderly, and increase the risk of fall injuries, causing a negative impact on the quality of life of the elderly [2]. Hence, the regular monitoring of the performance of ADLs in the elderly could support the independent daily living of the elderly and assist in maintaining their health [3]. Further, monitoring the performance of ADLs among the elderly is a high priority in terms of health management because it helps the doctor, family members, and caregiver to know the current health status of the elderly [4]. Monitoring the performance of ADLs is a process of continuous observation and digital recording of the daily living and physical balance ability of the elderly to determine the variation in the performance ADLs, which shows a high correlation with the variation in dynamic balance ability (DBA) in the elderly [5]. The DBA is defined as the ability to maintain and control the center of weight within the base of support when performing a physical movement [6]. As individuals age, they begin to display gradually reduced physical and psychological functions, which causes a decline in the DBA in the elderly [7,8]. It is thus crucial to assess the DBA when monitoring the performance of ADLs in the elderly.

To assess the DBA, monitoring- and devise-based evaluation methods have been used in clinical test fields. In monitoring-based methods, clinical evaluations of the DBA, such as with the four square step test (FSST), the short physical performance battery (SPPB), the timed up and go test (TUG) and the six-minute walk test (6MWT) are performed. The results are then recorded and interpreted by the clinical pathologist [9]. In a study by Cleary and Skornyakov [10], elderly individuals with a past experience of fall were screened based on the FSST, which, alongside the TUG test, was shown to be most effective in categorizing fall experience. Volpato et al. [11] assessed the SPPB in long-term hospitalization patients and performed interviews to predict balance ability, rehospitalization and the mortality rate, and the SPPB was reported to show a high explanatory power on the balance ability and rehospitalization of such patients. In Gadelha et al. [12], the TUG test was applied in community-dwelling female elderly with the aim of examining the risk of fall in female elderly individuals based on sarcopenia status, and the results showed that sarcopenia could increase the risk of falls in female elderly individuals. Caballer et al. [13] applied the 6MWT to identify the correlation of gait ability with lower-limb function, mobility and balance ability in the elderly, and the results showed that the gait distance in the 6MWT increased as lower-limb function and mobility increased. While monitoring-based methods are advantageous in simple assessment procedures and rapid screening of the DBA in the elderly, there is a risk of bias, as the subjective opinions of the rater could be reflected in the results. In addition, this approach may impose intense labor on nurses or clinical pathologists in this context of a growing elderly population [14].

In the device-based DBA evaluation methods, a professional physical kinematics device, such as the optoelectronic motion capture system (OMCS), the Kinect or the inertial measurement unit (IMU) is used to perform the clinical evaluation of DBA, and determine the DBA based on the collected data [15]. The OMCS and Kinect devices measure physical activities through vision with an outstanding level of accuracy; they are thus widely used across studies investigating the assessment of physical activities. However, the drawbacks include high spatial constraint depending on the venue of assessment, the high cost of the device, and the need for professional staff at all times to operate the device. Among the previous studies addressing these issues, Yorozu et al. [16] suggested a method of tracking the lower-limb activity upon the turn activity of the TUG test in the elderly through the use of the OMCS. Eltoukhy et al. [17] investigated the validity of the Kinect in the assessment of static and dynamic balance abilities, and for the single leg standing test in younger and older adults, the kinematics data of Kinect compared with the golden standard device showed a strong agreement.

The previously mentioned drawbacks of the OMCS or Kinect devices can be overcome through the use of the IMU, which is a small wearable sensor that can be attached to the body to measure the acceleration, the angular velocity and the direction of physical movements. In addition, the accuracy of IMU measurement has increased with the advancement of the micro electro mechanical systems (MEMS) technology, so that the device is frequently used in studies investigating the measurement of various physical activities [18,19,20]. The IMU-based TUG test was performed in Zakaria et al. [21], where the DBA was compared through the analysis of kinematics data collected via the IMU between the elderly and the general adult population. Lepetit et al. [22] suggested a method of quantifying the fall in the elderly based on the IMU data obtained in the 30-s sit-to-stand test. The device-based methods of DBA evaluation are advantageous in regard to their high levels of evaluation accuracy and reliability; nevertheless, the level of difficulty in interpreting the quantified result is high.

As the result of the device-based DBA evaluation is difficult for the elderly to understand and interpret, an expert opinion or additional explanation is necessary. Presenting the DBA status in the form of age or a score that can be intuitively perceived is likely to assist in the management of DBA in community-dwelling elderly [23]. This study thus aimed to generate a regression model based on machine learning (ML) to predict the age of the elderly based on the IMU data obtained in the DBA evaluations. The performance of the resulting model was also compared based on the IMU data of the TUG test and the 6MWT.

## 2. Materials and Methods

### 2.1. Participants

In this study, 136 subjects aged 60–90 years (mean age: 76.5 ± 7.0 years; height: 154.0 ± 8.1 cm; weight: 57.5 ± 9.3 kg) were recruited. Among these, 33 were males and 103 were females. In terms of age, 36 participants were in their 60s, 64 were in their 70s, and 36 were in their 80s. All participants voluntarily agreed to participate and informed consent was obtained from all subjects involved in this study. The inclusion criteria were as follows: aged 60 years or above; no history of musculoskeletal disorder in the past three months; no severe chronic illness; no history of mental disorder; no allergic reaction to sensor attachment; and the ability to understand the study contents and follow instructions independently. This study was approved by the Public Institutions Bioethics Committee designated by the Ministry of Health and Welfare of the Republic of Korea (P01-201902-13-001).

### 2.2. Data Collection

The purpose of data collection in this study was to predict the age of the elderly based on the kinematics data for physical activities recorded through the IMU. Only a single IMU (Research PRO, Noraxon, Scottsdale, AZ, USA) was used for data collection (Figure 1a). This IMU comprised a three-axis accelerometer, a three-axis gyroscope, and a magnetometer. The three-axis accelerometer can measure up to ±16G and its maximum data-sampling rate is 400 Hz. The three-axis gyroscope can measure up to ±2000 deg/s at a 400 Hz data sampling rate. The magnetometer was used to estimate the location and orientation of the sensor, but the magnetometer data did not be used to generate the model for predicting the age. The single IMU was attached to the second lumbar vertebrae using an elastic band (Figure 1b). Multiple IMUs could induce physical burden and discomfort in the elderly in an actual daily living environment [24,25]. Thus, a single IMU was used in this study, and the position of the IMU attachment was selected as the L2 close to the center of mass of body, based on previous studies [26]. The software used in IMU data collection was the MR 3.12 (Noraxon, USA). The data of the three-axis accelerometer and three-axis gyroscope were collected at a 100 Hz data sampling rate.

The kinematics data of physical activities of the elderly were collected using the TUG test and the 6MWT, two well-known clinical tests which allow a comprehensive assessment of the DBA in the elderly. In the TUG test, participants perform the following sequence of activities: sit to stand, forward gait, mid-turn, backward gait, end-turn and stand to sit [27]. The TUG test is widely used in clinical studies because of its simple procedures and the inclusion of a comprehensive set of activities closely associated with ADLs [7,28]. The 6MWT assesses the repeated gait on a 30 or 50 m linear distance within six minutes [29]. In the 6MWT, the total gait distance within six minutes is measured, and in addition to the gait distance, the data of various gait parameters including the gait speed, stride length and cadence are collected so that the overall mobility of the elderly can be assessed [30,31]. The 6MWT is used in various clinical studies regarding the test of physical functional performance in the elderly based on simple procedures, and because its reliability has been verified in a variety of studies [32,33].

The process of data collection is illustrated in Figure 2. First, the participants were provided an explanation of the study purpose and contents. The IMU was then attached to the L2 of the participant using an elastic band. The TUG test and the 6MWT were subsequently performed, the former preceding the latter. Prior to the tests, the participant was given time to practice the relevant activities for familiarization. The TUG test was performed in triplicate. At the onset, the investigator requested the participant to perform the TUG test as fast as possible. The chair used in the TUG test had a back but no armrest. The distance between the chair and the turn point was set to 3 m. The direction of turn at the turn point, or of rotation upon sitting on the chair were not restricted. Between each of the three trials, a 1 min rest was allowed. At the end of the last trial, the participant was allowed a 5 min rest prior to the 6MWT. For the 6MWT, the distance to the turn point for the 6MWT was set to 30 m. The participant was allowed to take a rest upon request throughout the 6MWT, according to the manual, and the investigator encouraged the participant to walk as far as possible [34]. For each participant, data collection lasted approximately 40 min.

### 2.3. Data Analysis

Figure 3 presents the framework for generating a regression model based on machine-learning (ML) to predict the age of the elderly. The framework consists of five stages; raw data extraction, data pre-processing and features engineering (performed using the MATLAB R2018a, MathWorks, Natick, MA, USA), model training, and validation (performed using Python 3.8, https://www.python.org/, accessed on 16 March 2022).

#### 2.3.1. Raw Data Extraction

The TUG test raw data included the acceleration and angular speed data. For the raw data of acceleration, the vertical linear acceleration (ACC_VT_), mediolateral linear acceleration (ACC_ML_), anterior–posterior linear acceleration (ACC_AP_), and the resultant acceleration (ACC_RES_) of the three-dimensional linear acceleration were used. The ACC_RES_ was obtained using the following equation:(1)$${\mathrm{ACC}}_{\mathrm{RES}}=\sqrt{{\mathrm{ACC}}_{\mathrm{VT}}{}^{2}+{\mathrm{ACC}}_{\mathrm{ML}}{}^{2}+{\mathrm{ACC}}_{\mathrm{AP}}{}^{2}}$$

For the raw data of the angular velocity, the vertical angular velocity (YAW), the mediolateral angular velocity (PITCH), and the anterior–posterior angular velocity (ROLL) were used. The 6MWT raw data contained only the acceleration data, for which the extraction method was identical to that in the TUG test.

#### 2.3.2. Data Pre-Processing

For the seven sets of raw data obtained in the TUG test, background noise was removed using the Butterworth low-pass filter (passband: 0.01π/sample; steepness: 0.83; stopband: 60 dB). The clean data were subsequently used in the analysis of the six sub-tasks in the TUG test. Each of these sub-tasks is an individual physical activity indicating the DBA; hence, each should be analyzed in its own right [35]. The six sub-tasks in the IMU data were differentiated in reference to the method suggested in Beyea et al. [36]. According to this method, rectification should be performed on the PITCH and YAW data from the IMU, after which the data noise can be removed through digital filtering, while the chair activity and the turn activity are distinguished based on the threshold (Figure 4). The motions of the chair activity are identified based on the PITCH (threshold: 25 deg/s), and those of the turn activity are identified based on the YAW (threshold: 32 deg/s). To determine the gait activity, the time between the chair activity and the turn activity is applied. For the four sets of raw data of acceleration in the 6MWT, data pre-processing involved noise removal using the Butterworth low-pass filter (passband: 0.02π/sample; steepness: 0.86; stopband: 40 dB).

#### 2.3.3. Features Engineering

Feature engineering process has played a vital role to generate the ML-based regression model. It is especially important that the feature was selected for improving the performance of the models. Amjad et al. [37] reported that handcrafted features performed quite well to recognize the ADLs. For this reason, 132 handcrafted features were extracted from the raw data collected in the TUG test and the 6MWT. A total of 132 handcrafted features were extracted from the raw data collected in the TUG test and the 6MWT. Among the 132 features, 111 were related to the TUG test, and 21 were related to the 6MWT. The TUG test features contained the features related to the time taken for the TUG test and descriptive statistics of seven signals (ACC_VT_, ACC_ML_, ACC_AP_, ACC_RES_, YAW, PITCH and ROLL) (Table 1). In previous studies investigating the DBA in the elderly using the IMU-based TUG test, RMS, min, and max were reported to most adequately represent the DBA in the elderly among descriptive statistics features [38,39]. For the features of the gait activity, the forward gait and backward gait were averaged.

The features of the 6MWT included the gait parameter (GP), the symmetry parameter (SP), the harmonic ratio (HR) and the approximate entropy (ApEn) (Table 2). The GT exhibits the spatiotemporal characteristics of gait. For the GT, the peak value of the ACC_VT_ data collected via the IMU during the 6MWT was used in step perception to calculate the total step count during the 6MWT. The mean step time was estimated using the step count to verify the time between each step. The height of the participant was used to estimate the stride length to obtain the gait distance during the 6MWT, and the gait speed was calculated using the estimated gait distance.

The GS was used to quantify the level of gait symmetry. The GS quantification involves step regularity (SR), stride regularity (STR), and the symmetry index (SI), which can be obtained through the autocorrelation function on the trunk location acceleration signal [40,41]. The SR is defined as the symmetry between each step, and the STR indicates the symmetry between strides. The SI is an indicator of the overall symmetry of the gait. The SR and STR can be derived from the coefficient in an unbiased-autocorrelation regarding vertical acceleration and anterior–posterior acceleration. The A_unbiased_ as the unbiased-autocorrelation function is calculated using the following equation:(2)$$A_{\mathrm{unbiased}}=\frac{1}{N-\left|m\right|}\overset{N-\left|m\right|}{\underset{i=1}{\sum }}x_{i}x_{i+m}$$
where N is the total number of data samples, m is the time lag parameter, x_i_ is the acceleration at time i, and x_i+m_ is the acceleration at time after applying m. The SR and STR can be determined based on the peak of A_unbiased_. The first peak of the A_unbiased_ coefficient following the Zero lag is defined as the SR, and the second peak is defined as the STR. The coefficient is a value between 0 and 1; values closer to 1 indicate a higher level of symmetry [42,43]. The SI was obtained by normalizing the max value between the SR and the STR.
(3)$$\mathrm{SI}=\frac{\left|\mathrm{SR}-\mathrm{STR}\right|}{\max\left(\mathrm{SR},\;\mathrm{STR}\right)}$$

The HR is used to evaluate the smoothness of the pattern of acceleration signals detected in gait [44]. The harmonic coefficients can be obtained through the Discrete Fourier Transform regarding the acceleration signals for ten strides. The HRs of ACC_VT_ and ACC_AP_ are calculated by dividing the sum of the even harmonic coefficients by the sum of the odd harmonic coefficients. In contrast, the HR of ACC_ML_ is calculated by dividing the sum of the odd harmonic coefficients by the sum of the even harmonic coefficients. The smoother the pattern of the acceleration signal, the higher the HR appears [45,46].

The ApEn is used to estimate the regularity of the acceleration signals during gait [47]. The ApEn is among the entropies indicating the complexity of time-series data, and the value of ApEn thus decreases as the regularity of the gait increases. The ApEn was derived using the method suggested in Pincus [48].

#### 2.3.4. ML-Based Regression

To generate the regression model to predict the age of the elderly, the feature data sets were divided into training, test, and validation sets. Of the total feature data sets (100%), 70% was given to the training set, 10 % to the validation set, and 20% to the test set. The training and validation sets were used for training and optimization of the regression model, respectively. The test set was used to evaluate the performance of the generated regression model.

To examine and optimize conventional supervised ML-based regression algorithms, a performance comparison of four algorithms was made. Selected four algorithms is following as: Random forest, XGBoost, Support vector machine (SVM) and Artificial neural net (ANN). These algorithms have been known that the performance was relatively high among traditional supervised ML-based regression algorithms in field to the human motion analysis. As a result of preliminary examination to select the most suitable the ML-based regression algorithm, the model using the XGBoost algorithm showed the most outstanding performance among four algorithms (Figure 5a,b).

The XGBoost algorithm was used as the algorithm of the ML-based regression model. The XGBoost algorithm is a boosting ensemble method based on the decision tree. This boosting method combines several shallow decision trees, which successively apply error-related weights [49]. The XGBoost algorithm allows parallel learning through gradient boosting so that enhanced performance and speedy learning can be anticipated [50]. Another function of the XGBoost algorithm is the verification of the importance of the feature used in model generation, with a consequent advantage of the ML outcome interpretation. The importance of each feature was determined by the feature importance score (FIS) function of the XGBoost algorithm. The FIS values range between 0 and 1, where those closer to 1 are regarded as having greater importance in the generation of the regression model. The hyperparameter tuning of the XGBoost algorithm was based on the Bayesian optimization. To prevent potential overestimation by the regression model in hyperparameter tuning, only the validation set was used.

To evaluate the performance of the regression model to predict the age of the elderly based on the DBA, a model was generated for each of the following: Only the TUG test features set (OT), only the 6MWT features set (OS), and the aggregated TUG test and 6MWT features set (AG), and the performance was compared across the models. For the comparison, the mean absolute error (MAE) and mean absolute percentage error (MAPE) were used as they are the representative indicators of regression model performance. The MAE allows the mean absolute error among the true and predicted values to be determined. The MAPE is the percentage form of the MAE that allows the error rate to be identified. The MAE and MAPE can be calculated as follows:(4)$$\mathrm{MAE}=\;\frac{1}{n}\sum \left|\hat{y}-y\right|$$
(5)$$\mathrm{MAPE}=\;\frac{100}{n}\sum \left|\frac{\hat{y}-y}{y}\right|$$
where n is the number of samples, $\hat{y}$ is the predicted value, and y is the actual value. For both MAE and MAPE, a smaller value indicates a greater level of model performance.

## 3. Results

The results of the regression models generated to predict the age of the elderly according to the DBA evaluations were as follows: the mean MAE and mean MAPE of OT, OS and AG models were 4.7 ± 0.8 years and 6.6 ± 1.2%, respectively. Figure 5 presents the results of the comparison of performance among the regression models. The AG exhibited the most outstanding performance (MAE = 3.8; MAPE = 5.2%), while the OT showed the lowest performance (MAE = 5.3; MAPE = 7.5%), with OS in between (MAE = 4.9; MAPE = 7.0%).

The top five most important features in the generation of each regression model are presented in Figure 6. Figure 6a shows the top five most important features among the 6MWT and TUG test features. For the 6MWT features, the average gait speed showed a high FIS in both AG and OS models, while the gait distance showed a high FIS in AG only, and the number of steps, mean step time, SR of ACC_VT_ and STR of ACC_RES_ showed a high FIS in OS only. For the TUG features, the features related to the turn activity showed a high FIS. To be specific, the RMS of YAW for end-turn and the minimum of YAW for end-turn showed a high FIS in both AG and OT; the RMS of ACC_ML_ for end-turn showed a high FIS in AG only; the RMS of ACC_ML_ for mid-turn, the maximum of ACC_ML_ for mid-turn and the maximum of ACC_ML_ for end-turn showed a high FIS in OT only. Figure 6b–d show the FIS of the top five most important features in the model generation. First, the AG showed a high FIS for the ACC_RES_ RMS for the end-turn (0.046), the average gait speed (0.046), total gait distance (0.039), the minimum YAW for the end-turn (0.036), and YAW RMS for the mid-turn (Figure 6b); the OS showed a high FIS for the STR (0.121) of ACC_RES_, the SR (0.118) of ACC_VT_, the average gait speed (0.113), the number of step (0.096) and the average step time (0.093) (Figure 6c); the OT showed a high FIS for the minimum YAW for the end-turn (0.116), the maximum ACC_ML_ for the mid-turn (0.090), the YAW RMS for the mid-turn (0.090) and the ACC_ML_ RMS for the mid-turn (0.090), and the maximum ACC_ML_ for the end-turn (0.080) (Figure 6d).

## 4. Discussion

In this study, an ML-based regression model to predict the age of the elderly was generated to monitor the performance of ADLs in the elderly. Furthermore, the performance of the resulting model was analyzed through comparison using the feature data sets of the TUG test and the 6MWT in DBA evaluations. The IMU data obtained in the TUG test and the 6MWT by the elderly aged 60 years or above were used in the regression model to predict the age of the elderly. To enhance the performance of the regression model, both the features of description statistics, and the DBA-related features reported in previous studies were taken into account.

Among the three regression models to predict the age of the elderly using the TUG test and 6MWT feature data sets, the model incorporating all of the TUG test and 6MWT features showed the most outstanding performance, presumably because the explanatory power on the variation in DBA with the increase in age was improved when the performance of ADLs and the gait ability were tested simultaneously. In the study by Steffen et al. [51], the age of the elderly showed strong correlations with the TUG test and 6MWT results, prompting the authors to suggest that both tests be performed and their results interpreted together when testing the DBA in the elderly. Caballer et al. [13] reported a significant correlation between the TUG test and the 6MWT in the assessment of physical performance in community-dwelling elderly, with both tests having high levels of reliability.

The regression model proposed in this study simultaneously applies the features of the TUG test and the 6MWT, and showed that the features related to the turn activity (RMS of ACC_RES_ for end-turn, minimum of YAW for end-turn and RMS of YAW for mid-turn) and the GP (average gait speed and gait distance) that influence the prediction of age in the elderly according to the respective DBA evaluations had a positive effect on the model performance. In the model incorporating the TUG test features only, a high FIS was displayed by the features related to the turn activity (minimum of YAW for end-turn, maximum of ACC_ML_ for mid-turn, RMS of YAW for mid-turn, RMS of ACC_ML_ for mid-turn and maximum of ACC_ML_ for end-turn), which indicates that the ability of turn activity could change according to the age of the individual. This result agrees with that of Dite and Temple [52], who reported a higher risk of fall in the elderly with a lower performance of turn activity, as well as that of Nordin et al. [53], who reported a lower performance of turn activity in the TUG test in the elderly of further progression in frailty. The results collectively supported the difficulty felt by the elderly with increasing age in performing the motions related to the turn activity in daily living due to decreased the DBA.

In the model incorporating the 6MWT features only, a high FIS was displayed by the features related to the GP (average gait speed, step count and step time) and the GS (STR of ACC_RES_ and SR of ACC_VT_). Previous clinical studies on the 6MWT have shown that the GP (average gait speed, step count and step time) was a critical indicator of the frailty in the elderly, which would for the results in this study [54,55]. Interestingly, in agreement with previous studies, the features related to the GS (STR of ACC_RES_ and SR of ACC_VT_) showed a high FIS. Kobayashi et al. [41] reported the low SR and STP of ACC_VT_ for the individuals in their 70s compared to individuals in their 20s. Martínez-Ramírez et al. [43] reported low SR and STP of ACC_VT_ for those elderly individuals with a greater increase in frailty compared to the general elderly population. Thus, as with previous studies, the results in this study showed that the asymmetrical gait acceleration pattern in the elderly could be attributed to the decline of DBA with increasing age.

Additionally, the average gait speed and the total gait distance exhibited a high FIS in the model incorporating both the TUG test and 6MWT features. This is presumed to be because the two features are the most commonly used ones in the assessment of the gait ability in the elderly in clinical studies applying the 6MWT.

As we mentioned in the Introduction, the conventional method to determine the DBA status required the elderly to visit the hospital and the clinical pathologists opinion was necessary. As this work represented the DBA status in the form of age that can be easily perceived by the elderly, the elderly can monitor their own DBA status by myself at home and this may be effective in preventing injuries caused by falls. This study has two significant limitations which should be pointed out. First, in the collection of IMU data of the TUG test and the 6MWT to predict the age of the elderly, the number of samples in each age group was not identical. The number of samples collected from the individuals in their 70s was the greatest, which implicates a possibility of bias in the model towards this age group, and corresponding model validation is required. In a follow-up study, therefore, the same number of samples will be collected from individuals in their 60s and 80s as with those in their 70s for complementation. Finally, the model will be further advanced and finalized by collecting data from the general adult population in their 20s–50s. Second, gender-specific characteristics were not considered in the generation of the regression model to predict the age of the elderly in this study. As it is possible that the variation in DBA caused by aging is unique to each gender, the model will be complemented with respect to gender characteristics in a follow-up study.

## 5. Conclusions

In this study, we generated an ML-based regression model to predict the age of the elderly. For this, the IMU data of the TUG test and the 6MWT were collected from the elderly participants and various features with high correlations with the DBA in the elderly were determined by feature engineering. In addition, the model performance was comparatively analyzed using the TUG test and 6MWT feature sets, with the model based on all of the TUG test and the 6MWT features showing the most outstanding performance. We were thus able to prove the usefulness of the indicator of age in monitoring the performance of ADLs in the elderly. The findings in this study provide the basic data in the IMU-based quantification of the DBA in the elderly and in the development of ML-based DBA monitoring system for the elderly.

## Figures and Tables

**Figure 1 sensors-22-05957-f001:**
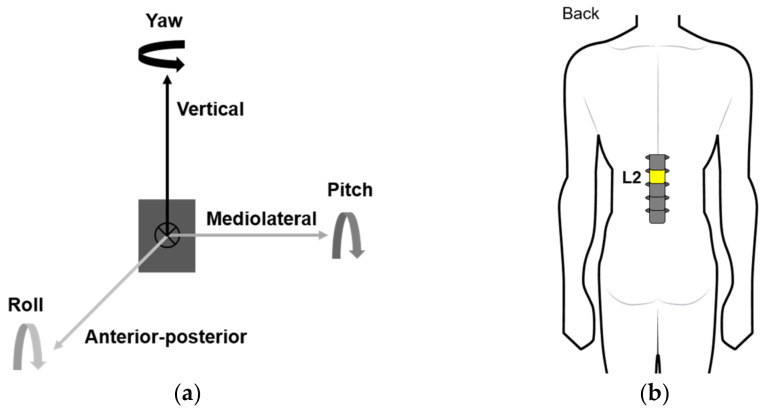
Information of the IMU: (**a**) definition 3-axis of the IMU; (**b**) attachment position of the IMU.

**Figure 2 sensors-22-05957-f002:**
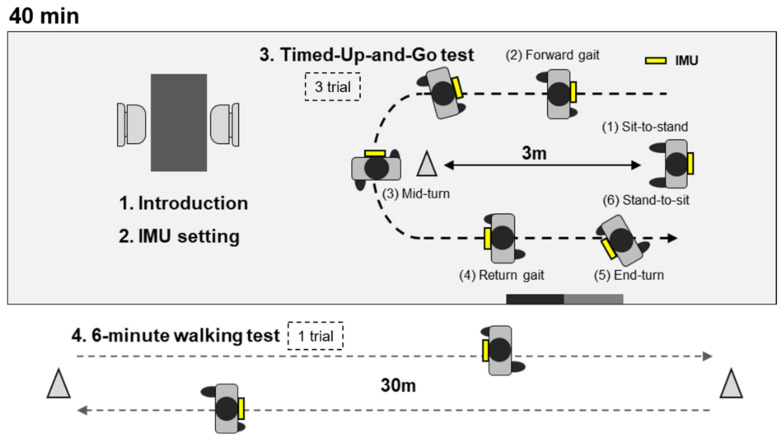
Data collection procedure and environment.

**Figure 3 sensors-22-05957-f003:**
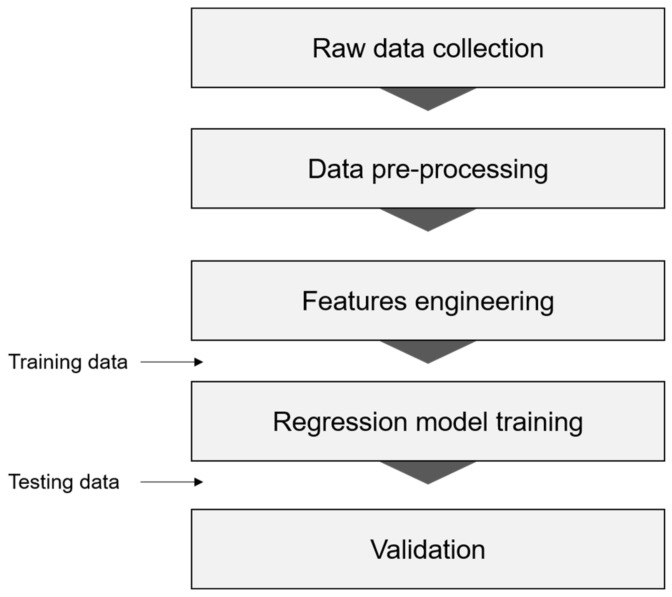
The framework for generating the ML-based regression model.

**Figure 4 sensors-22-05957-f004:**
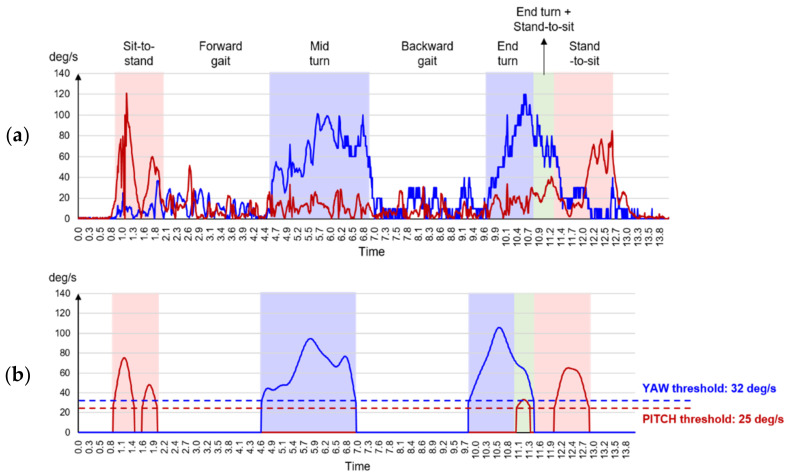
Example of identification of the TUG test sub-tasks (red line: PITCH signal; blue line: YAW signal): (**a**) rectification for raw data; (**b**) results of the TUG test data pre-processing for recognizing the TUG test sub-tasks.

**Figure 5 sensors-22-05957-f005:**
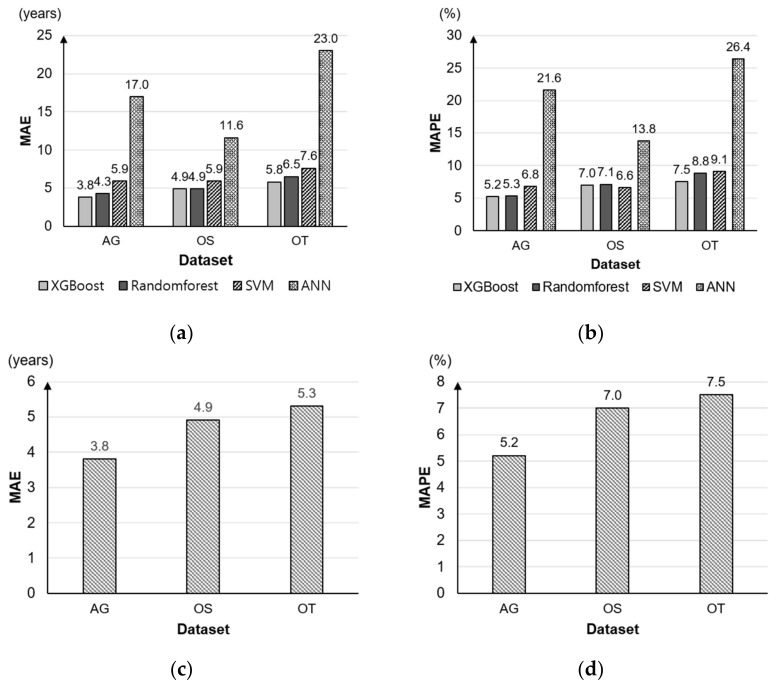
Performance comparison of the regression models: (**a**) MAE comparison results between four algorithms; (**b**) MAPE comparison results between four algorithms; (**c**) MAE results of XGBoost models to each datasets; (**d**) MAPE results of XGBoost models to each datasets.

**Figure 6 sensors-22-05957-f006:**
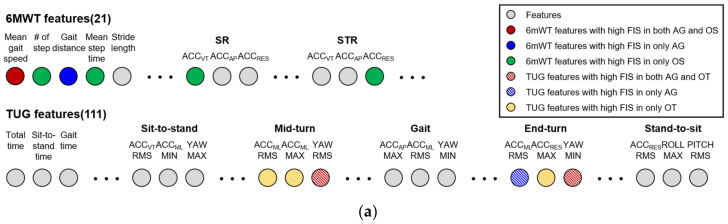
Analysis of feature importance in each of the XGBoost models as the dataset: (**a**) key features for each XGBoost model as the dataset; (**b**) AG; (**c**) OS; (**d**) OT.

**Table 1 sensors-22-05957-t001:** Definitions of TUG test features.

	Features	Definition
Time features	Total time	Total time on the TUG test
	Sit-to-stand time	Time from sit on a chair to stand
	Gait time	Average time on forward gait and backward gait in the TUG test
	Mid-turn time	Time on rotation at return point
	End-turn time	Time on rotation for sit on a chair
	Stand-to-sit time	Time from stand to sit on a chair
Descriptive statistics features	Root mean square (RMS)	Arithmetic mean of the squares of a set of values
	Min	The smallest value
	Max	The greatest value

**Table 2 sensors-22-05957-t002:** Definitions of 6MWT features.

	Features	Definition
GP	Number of steps	Number of steps taken during 6 min
	Step/s	Step per second
	Step time	Mean time between each step
	Stride length	Distance between steps
	Gait distance	Walking distance for 6 min
	Average gait speed	Average walking speed for 6 min
GS	Step regularity	Symmetry between steps as identified by ACC_VT_, ACC_AP_, ACC_RES_ for walking
	Stride regularity	Symmetry between strides as identified by ACC_VT_, ACC_AP_, ACC_RES_ for walking
	Symmetry index	Gait symmetry index
HR	Harmonic ratio	Smoothness of acceleration signals measured for walking
ApEn	Approximate entropy	Regularity of acceleration signals measured for walking

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
