# Peer review of "Machine Learning-Based Predicted Age of the Elderly on the Instrumented Timed Up and Go Test and Six-Minute Walk Test"

_sensors, 2022, doi:10.3390/s22165957_

Round 1

Reviewer 1 Report

The authors present a regression model based on machine learning to predict the age of the elderly. The model was based on inertial measurement unit data as part of the DBA evaluation, and the performance of the model was comparatively analyzed with respect to age prediction based on the IMU data of TUG and 6MWT.  The XGBoost algorithm was used to train the regression model. Comparison of the regression model performance according to the TUG and 6MWT feature sets showed that the performance was best for the model using all features of the TUG and 6MWT.

The article is well written, nevertheless, some English spell check is required. In order to help to improve the article, some comments are suggested as follows.

1.      The discussion of previous work dealing to predict the age of the elderly based on machine learning is still weak and needs to be better highlighted.

2.      The algorithm used for the prediction is XGBoost. The choice of this method is not justified nor compared with other techniques to prove this choice.

3.      I propose to present a scenario containing the observations of an elderly and the result of the prediction from the used ML model.

4.      Among the 50 references, there are only two dating from 2020. I propose to reference some recent works from the last two years. Here are some suggestions from recent work that you can cite in case you think they relate to your work:

-       Hu, R.; Michel, B.; Russo, D.; Mora, N.; Matrella, G.; Ciampolini, P.; Cocchi, F.; Montanari, E.; Nunziata, S.; Brunschwiler, T. An Unsupervised Behavioral Modeling and Alerting System Based on Passive Sensing for Elderly Care. Future Internet 202113, 6. https://doi.org/10.3390/fi13010006

-       D’Onofrio, G.; Fiorini, L.; Toccafondi, L.; Rovini, E.; Russo, S.; Ciccone, F.; Giuliani, F.; Sancarlo, D.; Cavallo, F. Pilots for Healthy and Active Ageing (PHArA-ON) Project: Definition of New Technological Solutions for Older People in Italian Pilot Sites Based on Elicited User Needs. Sensors 202222, 163. https://doi.org/10.3390/s22010163

-       Aloulou, H.; Mokhtari, M.; Abdulrazak, B. Pilot Site Deployment of an IoT Solution for Older Adults’ Early Behavior Change Detection. Sensors 202020, 1888. https://doi.org/10.3390/s20071888

-       Guduru, R. K. R., Domeika, A., Dubosiene, M., & Kazlauskiene, K. (2021). Prediction framework for upper body sedentary working behaviour by using deep learning and machine learning techniques. Soft Computing, 1-16.

-       Petrushin, A., Freddolini, M., Barresi, G., Bustreo, M., Laffranchi, M., Bue, A. D., & Michieli, L. D. (2022). IoT-Powered Monitoring Systems for Geriatric Healthcare: Overview. Internet of Things for Human-Centered Design, 99-122.

5.      A few minor spell check should be corrected. For example:

Page 7: The Aunbiased as the unbiased autocorrelation function is calculated using the following equation: à Aunbiased (unbiased as an index)

Reviewer 2 Report

This paper presents a technique to  predict the age of elderly people based on the IMU data of daily activities using machine learning techniques. The data of physical activities is collected during two well-known clinical tests: TUG 133 and 6MWT. The proposed technique extracts a few very simple handcrafted features from the sensory data i.e.,  min, max, root mean square, number of steps. step time, stride length ad etc. to predict the age of elderly using machine learning-based regression model named as, XGBoost algorithm.

 In my point of view, the following concerns should be addressed:

·         The proposed methodology is lacking with novelty. The authors employed some standard set of handcrafted features and a well-known machine learning tool to estimate the age of elderly people. These features have been used in several paper from last many years to solve the problem of movement analysis in the domain of healthcare. The scientific contribution must be stated clearly.

·         An IMU sensor consists of accelerometers (to measure acceleration), gyroscopes (to measure angular rate), and magnetometers (to measure magnetic field surrounding the system). The authors only employed accelerometers and gyroscopes in their data collection however, they did not mention any scientific reason to either including both accelerometers and gyroscopes or excluding magnetometers sensors. A comparative study shall be included in the paper to clearly state the reason of using these two sensors.

·         Rather than using very simple feature encoding technique; I would like to see the evaluation of 16 handcrafted features proposed in [1].

[1]  F. Amjad, M.H. Khan, M.A. Nisar, M.S. Farid, M. Grzegorzek, “A comparative  study  of feature selection approaches for human activity recognition using multimodal sensory data, ” Sensors, vol. 21, 2021 

·         The authors should publicly available the collected dataset to the research community and this may be treated as a strong contribution.

·         Wearing a set of IMU on human-body can minimize the actual movement of the subject (especially to elderly people) to some extent and discomfort in their actual daily living activities. Why the authors did not explore the same sensory data from the devices of daily usage e.g., Mobile device. For reference see [2]. 

[2]  Köping, Lukas, Kimiaki Shirahama, and Marcin Grzegorzek. "A general framework for sensor-based human activity recognition." Computers in biology and medicine 95 (2018): 248-260.

·         The authors used Butterworth low-pass filter to remove the noise however, they did not mentioned that what type of noise they observed in the data and the relevance of this filter in the removal of noise.

·         The experimental evaluation is very limited. The computed results must compare with state-of-the-art techniques.

Round 2

Reviewer 1 Report

I would like to thank the authors for carefully addressing my comments. yet, I think they should provide a replication package of their work so that the community can reproduce and benefit from their model.

Reviewer 2 Report

Since the results of [1] have already been computed as commented in the first review cycle, I would like to suggest the authors to report the results of these features [1] in the revised manuscript.

[1] F. Amjad, M.H. Khan, M.A. Nisar, M.S. Farid, M. Grzegorzek, “A comparative study of feature selection
approaches for human activity recognition using multimodal sensory data, ” Sensors, vol. 21, 2021
